

# Metric-based analysis of the historical drivers of surface hydrological connectivity

Marta Antonelli[1], Victoria Scherelis[2,3], Christine Weber[1]

[1]Department of Surface Waters Research and Management, Swiss Federal Institute of Aquatic Science and Technology
(Eawag), Seestrasse 79, Kastanienbaum, CH-6047, Switzerland
[2]Institute of Natural Resource Sciences, Zurich University of Applied Sciences (ZHAW), Grüentalstrasse 14, Wädenswil,
8820, Switzerland
[3] Department of Geography, University of Zurich (UZH), Winterthurerstrasse 190, Zurich, 8057, Switzerland

*Correspondence to*: Marta Antonelli (marta.antonelli@eawag.ch)

**Abstract.** Hydrological connectivity is essential for the maintenance of important hydrological and ecological processes of catchments. Over time, human activities have altered the natural patterns of hydrological connectivity, leading to habitat loss and deterioration. Historical information from cartographic maps can be used to enhance our understanding of large-scale hydrological processes such as connectivity, by offering snapshots of past, less human-impacted landscapes and hydrological

systems. The focus of this study is on historical surface hydrological connectivity and its landscape drivers (e.g., lithology, topography, land use/ land cover) in ten Swiss catchments from different biogeographic regions (i.e., Pre-alpine, Alpine, Karstic, Plateau), and with varying physiographic characteristics. We employed hydromorphological metrics derived from historical maps (~ late 19th century) as proxies of surface hydrological connectivity, with the main goal of identifying the primary landscape drivers of connectivity. As expected from theory, the historical patterns of hydrological connectivity in

the studied catchments were mostly driven by landscape topography, and in particular by the slope and the morphology of the valley bottom. Unexpected relationships between connectivity and its drivers could be traced back to human practices, such as specific irrigation techniques and peat digging. Overall, our study shows how historical information can be employed to gain a deeper understanding on important large-scale hydrological processes, their primary drivers and on the history of human exploitation of the territory. Finally, this kind of approach paves the way for the characterization of how connectivity

has changed through time.



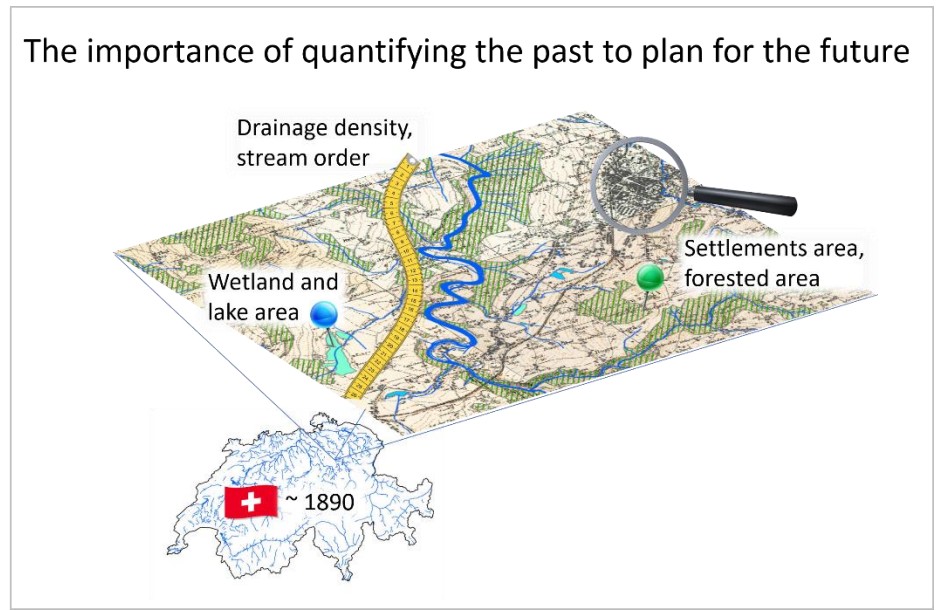

## 1. Introduction

Hydrological connectivity, defined as the "water-mediated transfer of matter, energy, and organisms within or between elements of the hydrological cycle" (Pringle, 2001), has emerged as a critical concept in understanding streamflow generation and sediment transport within catchments (Bracken and Croke, 2007; Parsons et al., 2015; Wainwright et al., 2011). Hydrological connectivity is a crucial property of the catchment, fundamental for the preservation of its ecological integrity. Because of their ecological relevance, a good understanding of the spatio-temporal patterns of hydrological connectivity – and disconnectivity – within a catchment is imperative to unravel the response of aquatic ecosystems to natural and anthropogenic disturbance. In fact, connection and disconnection between and within different sub-catchments strongly affect their resistance and resilience to perturbation (Wohl, 2017), for example by promoting species recolonization or by avoiding the spread of contaminants (Crook et al., 2015). For this reason, most restoration and conservation practices aim at recovering – or preserving – functional patterns of connectivity and disconnectivity among the different sub-catchments (Wohl, 2017; Fryirs, 2013).

The spatio-temporal patterns of connectivity within a catchment are strongly driven by the structural configuration of the landscape, including its geological and topographic setting, and its vegetation cover (Jencso et al., 2009; Van Nieuwenhuyse et al., 2011; Wohl et al., 2019). These heterogeneous landscape characteristics represent natural elements of (dis)continuity (Benda et al., 2004a; Poole, 2002), which influence the water movement across different dimensions of the hydrographic network (e.g. longitudinal, lateral and vertical dimensions; Brierley et al., 2006; Lexartza-Artza and Wainwright, 2011; Turnbull et al., 2018; Ward, 1989). However, in the course of history, landscapes and aquatic ecosystems have been increasingly exploited and modified by a variety of human activities, such as farming and urbanization, and the construction





of dams and other physical barriers (Zhang et al., 2021). This has led to unprecedented levels of alteration of the natural patterns of hydrological connectivity, with consequential deterioration and loss of the ecosystems dependent on them (Belletti et al., 2020; Grizzetti et al., 2017).

In today's highly anthropised landscapes, the primary drivers of fundamental catchment's functions such as connectivity are likely to have shifted from environmental to human factors (Allan, 2004; Allan et al., 2021). Resorting to historical information can provide essential insights into the past spatial distribution and primary drivers of connectivity, before the landscape and the hydrographic networks were subjected to severe modification and fragmentation (i.e., urbanisation, dam construction). The historical perspective is crucial for gaining a holistic view on today's catchments functioning, guiding

modern restoration and conservation efforts, and assessing the outcome of current actions (Higgs et al., 2014). For instance, historical connectivity patterns can serve as a basis for developing a benchmark for implementing and monitoring river restoration projects (Mould and Fryirs, 2018; Wohl et al., 2015). They can also aid in identifying near-natural water bodies and catchments, thus informing conservation prioritization strategies and ensuring efforts are grounded in the natural range of variability of the landscape (Speed et al., 2016; Grabska-Szwagrzyk et al., 2024).

In this contribution, we investigate the historical spatial patterns of surface hydrological connectivity in ten Swiss catchments characterised by different physical, geographical, and climatic characteristics. We rely on Switzerland's historical Siegfried map (developed between 1870 and 1926) to derive a series of hydromorphological metrics (cf. Scherelis et al., 2023), and employ them as a proxy of surface hydrological connectivity (SHC) (Brierley et al., 2006; Van Nieuwenhuyse et al., 2011). This approach allows us to characterise the primary drivers on surface hydrological connectivity

and, in particular, to I) understand if catchments with similar landscape characteristics (e.g., lithology, topography, land use/ land cover) show similar levels of surface hydrological connectivity, and to II) identify possible discontinuity in the expected patterns of connectivity, intended here as observed deviations from the hypothesized relationships between landscape drivers and hydrological connectivity (Table 1).

**Table 1: Hypothesized relationships between landscape drivers (i.e., lithology, topography and land use/land cover) and the spatial patterns of surface hydrological connectivity (SHC) in continental- temperate climate.**

| Landscape characteristic: | Hypotheses related to the distribution of SHC: | Reference |
|---|---|---|
| Lithology | • Catchments with a high percentage of permeable rock types present a low degree of longitudinal and lateral SHC. | Cotton, 1964; Sangireddy and others, 2016; Allan and others, 2021 |
| Topography | • Catchments with a high elevation, steeper slopes and a narrow valley width present a high degree of longitudinal and lateral SHC and are characterized by low stream orders<br>• Catchments with a small percentage of area covered by the valley bottom present a low degree of | Tucker and Bras, 1998; Benda and others, 2004b; Lin and Oguchi, 2004 ; Rice and others, 2008; Sangireddy and others, 2016; Allan and others, 2021 |





| | | |
|---|---|---|
| | longitudinal and lateral SHC | |
| Land use and land cover (LULC) | • Catchments with a high percentage of area covered by forest present a low degree of longitudinal and lateral SHC<br>• Catchments with a high percentage of area covered by buildings present low longitudinal and lateral SHC<br>• Catchments with a high percentage of area covered by lakes present low longitudinal SHC. However, lateral SHC may be promoted to some extent<br>• Catchments with a high percentage of area covered by wetlands present a high degree of longitudinal and lateral SHC. However, it may have a negative impact on the development of high stream orders<br>• Catchments with a high percentage of area covered by glaciers present a low degree of longitudinal and lateral SHC | Fergus and others, 2017; López-Vicente and others, 2017; Epting and others, 2018; Allan and others, 2021 |

## 2. Materials and methods

### 2.1 Historical source: the Siegfried map

The Siegfried map has been developed between 1870 and 1926 and covers the whole Swiss territory. It was the first Swiss map to depict features in colors (e.g., blue for hydrographic network, green for vegetation such as forested areas, black for buildings), and provides a unique opportunity for characterizing historical hydrographic networks.

The 604 map sheets that compose the Siegfried map are at two different scales depending on the location: the Jura, Central Plateau, and southern Ticino present a scale of 1:25000 (462 map sheets), while all the Alpine regions present a scale of

1:50000 (142 map sheets). The topographers responsible for the creation of the Siegfried map were required to adhere to specific instructions in order to guarantee a consistent delineation of the hydrographic network. The accuracy of the maps is relatively high, with a maximum acceptable error of 12.5 m and 35m for the areas with a scale of 1:25000 and 1:50000, respectively (Swisstopo).

The following features were extracted from the scanned map through machine-learning algorithms and translated into digital

vector data (see Wu et al., 2022 for more details): the hydrographic network (rivers, streams, lakes and wetlands), the forested areas, the area of the glaciers and the buildings' footprint (i.e. the area on the map covered by the single buildings; Heitzler and Hurni, 2020). Given the absence of a differentiation in terms of symbology between perennial and ephemeral streams, we assumed that the streams depicted on the Siegfried map represent the perennial component of the hydrographic



network. This is related to the fact that perennial channels may have been more likely to be consistently detectable for mapping in comparison to ephemeral channels. The features can be employed for the derivation of hydromorphological metrics of SHC, referred to as "connectivity metrics" or "metrics" from now on (see paragraph 2.2).

To cover the entire catchment area of the ten study catchments, we relied on the oldest available map sheets for each catchment, resulting in a time span of approximately 40 years between the oldest (dated 1872) and the newest map sheet (dated 1915). While this time span may seem extensive, it is crucial to note that not all map sheets were regularly updated, and updates were carried out in no apparent order (Räth et al., 2023). Consequently, landscape modifications might only appear on updated map sheets years after they occurred. Thus, we interpret the information from these various map sheets as providing an averaged snapshot of the spatial distribution of surface connectivity in the different catchments during the late 19th century.

## 2.2 Hydromorphological metrics of surface hydrological connectivity

Six hydromorphological metrics of SHC and their derivation process are presented in Fig. 1 and Table 2. The last two columns in Table 2 provide an overview of the metrics' function as a proxy of SHC and of their associated ecological relevance. The metrics of drainage and confluence density are here used as proxies of longitudinal surface connectivity. The spatial variability of these metrics provides important information on the overall wetness level of a catchment, its hydrological responses and structure (Benda et al., 2004b; Tucker et al., 2001). The metrics related to the stream order, such as the maximum stream order within each sub-catchment and the most frequent one (mode), are also used as proxy for longitudinal surface connectivity. However, compared to drainage and confluence density, the information they carry is more related to the hierarchical structure of the network within the catchments and the hydrological and morphological characteristics of the reaches (Huang et al., 2007; Leopold, 2006). Metrics related to the rivers and streams shoreline are here employed as a proxy for lateral connectivity. These metrics carry information on the physical extent of the shoreline which is available for lateral exchanges (i.e., shoreline density). By differentiating among different shoreline interfaces, we identify areas of generally high lateral connectivity (high percentages of shoreline bordering wetlands; Epting et al., 2018; Schmadel et al., 2019) and low lateral connectivity (e.g., high percentages of shoreline bordering buildings; Blanton and Marcus, 2009). High percentages of shoreline bordering forest can also be associated to low lateral connectivity (López-Vicente et al., 2017). All the metrics have been derived exclusively from the features extracted from the historical Siegfried map using ArcGIS Pro (Esri Inc.) and QGis (Qgis.Org).

Historical maps – or maps in general – come with inherent sources of uncertainty (Leyk and Boesch, 2009), which is often due to the scanning of the paper maps, image compression processes, as well as general aging and bleaching effects (Chiang et al., 2014b). Furthermore, simplifications and assumptions of features, inconsistences in map symbology, use of mixed colouring, and varying dimensions of map symbols due to the manual production techniques of the maps lead to additional uncertainties when extracting features from the maps (Lausch et al., 2015). As the metrics are derived from these extracted features, we should acknowledge that this uncertainty can affect our metrics. However, some forms of error may be





prohibitive for meaningful computations of certain metrics and be irrelevant for others (Scherelis et al., 2023). Since the metrics employed in this study are particularly sensitive to topology errors (i.e. erroneous relationships between the elements of the network and with the surrounding landscape; Leyk et al., 2005), we carried out the manual correction of gaps before

deriving our metrics (e.g., gaps between map-sheet edges, gaps within the hydrographic network emerging from the presence of bridges). Despite their inherent quality issues, historical maps offer a great opportunity to study historical landscapes and their changes in pre-digital times (Chiang et al., 2014a).

To assess to which extend the hydrographic network depicted on the Siegfried map provides a plausible representation of the real historical network, we calculated to which proportion the historical hydrographic network aligns with the valley bottom

(as defined and calculated in Sechu et al., 2021) as determined from contemporary DEM data (truth). In other words, assuming that the topography of the studied catchments has not changed, we expect the hydrographic network to be situated within the valley bottom. This provides an estimation of the areas within each catchment that are likely to provide more or less plausible metrics.

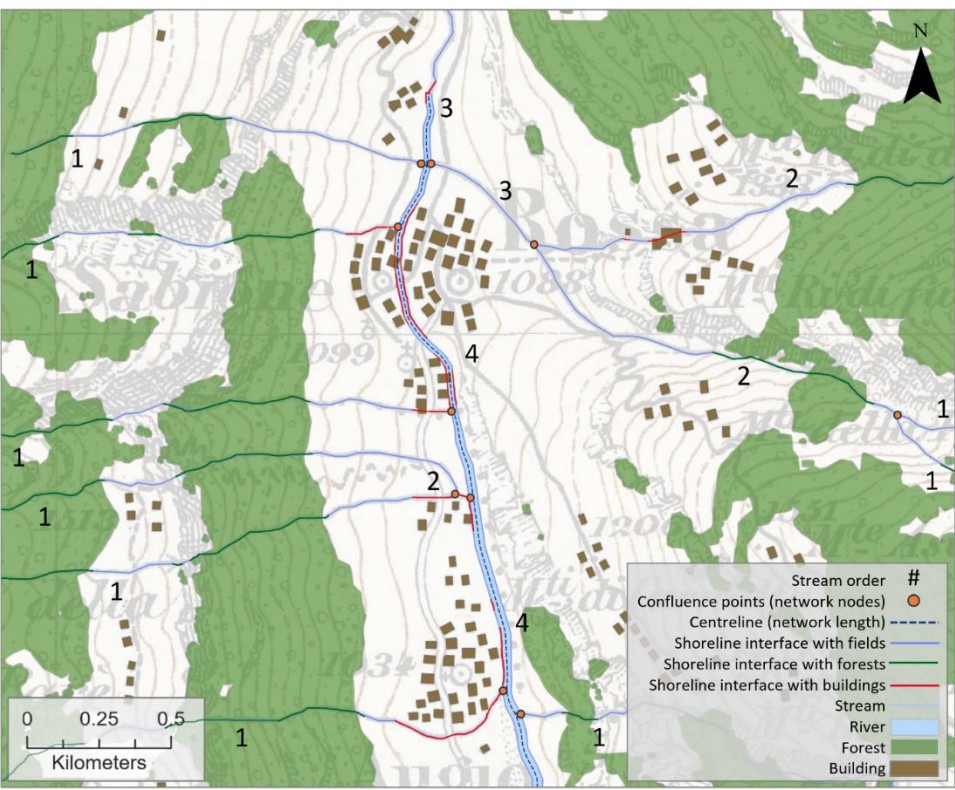


**Figure 1: Examples of feature appearance and metrics derivation (not exhaustive) shown on part of the Calancasca catchment (Siegfried map is visible in the background). Characteristics such as network length (here only shown for the river for the sake of clarity but calculated for the whole network) and confluence nodes form the foundation to derive the metrics, such as for drainage density or confluence density. Shoreline interface with different landscape features such as forest of buildings refer to the**
**proximity of the hydrographic network to these features (see also Table 2).**





**Table 2: List of the hydromorphological metrics of surface hydrological connectivity derived for this study.**

| | Metric | Definition | Derivation process | Proxy for surface hydrological connectivity | | Ecological relevance | |
|---|---|---|---|---|---|---|---|
| | | | | Structural | Functional | Structural | Functional |
| Longitudinal surface connectivity | Drainage density (1/km) | The sum of the channel lengths (rivers and streams) per unit area, a measurement that relates to the efficiency by which water is carried over the landscape | Total length of the network measured from the centreline of the rivers and streams per unit area | Physical connection within the sub-catchment, network complexity, overall sub-catchment wetness | Transport of water, sediment | | Transport of nutrients and organisms, species dispersion and recolonization |
| | Confluence density (N/km2) | The number of confluences (sections where flowing water bodies intersect) per unit area | Number of confluences (all intersecting points) of rivers and streams per unit area | Physical connection within the sub-catchment, network complexity, net morphological effect of confluences | Tributary effect (material and sediment input) | Habitat provision and complexity | Transport of nutrients and organisms, species dispersion and recolonization, network-scale community stability |
| | Stream order (max) (N) | Maximum Strahler order (Strahler, 1957) in the sub-catchment | Indicates the stream order number of the water body based on the flow direction of the river and the upstream value | Hierarchical location, placement of a certain catchment within the network, connection with the other catchments. Level of branching in a river system. | | How much the communities in a reach are dependent on/ influenced by the hierarchy of tributaries | Community type |
| | Stream order (mode) (N) | Most frequent Strahler order in the catchment | Same as for stream order (max) and filtering for value frequency instead of max stream order value | | Most common stream hydrological behaviour | | Community type |
| Lateral surface connectivity | Shoreline density (1/km) | The length of the shoreline divided by the surface area unit to evaluate and compare shoreline-associated impacts | Total length of the shoreline (rivers and streams) per unit area | Extend of the physical connection between the network (rivers and streams) and the terrestrial environment | Input/ exchange of water, sediment | Habitat provision and complexity | Input/ exchange of nutrients, organisms |
| | Shoreline interface forest, wetlands, buildings, fields (%) | Length of the shoreline (rivers and streams) which borders with different landscape features, namely forested areas, wetlands, urban settlements and single buildings, and fields (i.e., areas that are not included in the other categories) | Proximity of individual buildings, urban settlements (as defined and derived in Räth et al., 2023), wetlands, forests, and fields to the shoreline of rivers and streams. Percentages of shoreline interfaces. | Extend of the physical connection between the network (rivers and streams) and different categories of terrestrial (e.g., forest, buildings, fields) and semi-aquatic (e.g., wetlands) environments | Heterogeneity of the input/ exchange of water, sediment | Habitat provision and complexity | Heterogeneity of the input/ exchange of nutrients, organisms |

## 2.3 Catchment selection and characterisation

The ten Swiss catchments selected for this study are shown in Fig. 2. The catchments have been selected in order to represent the variability of the territory in terms of different physical, geographical and climatic characteristics (Table 3). Two paired catchments were selected for different biogeographical regions (i.e., regions presenting homogeneous



characteristics in terms of flora and fauna, Gonseth et al., 2001). Although each catchment is unique, they function as quasi-replicates for the metrics derivation, providing a comparative framework for the study.

We derived the connectivity metrics for the different catchments at different levels of their nested spatial structure: catchment, large sub-catchments (based on Federal Office of the Environment's topographical catchment areas of Swiss waterbodies 40 km²; Foen, 2024a), and small sub-catchments (based on Federal Office of the Environment's topographical catchment areas of Swiss waterbodies 2 km²; Foen, 2024b). The boundaries of both the large and the small sub-catchments have been slightly adapted to fit the generally more complex hydrographic networks showed on the Siegfried maps. We

focus on the metrics derived at the small sub-catchment level as our main dataset for the spatial analysis of connectivity. Finally, we calculated a set of landscape variables for each of the catchment's levels (Table 4).

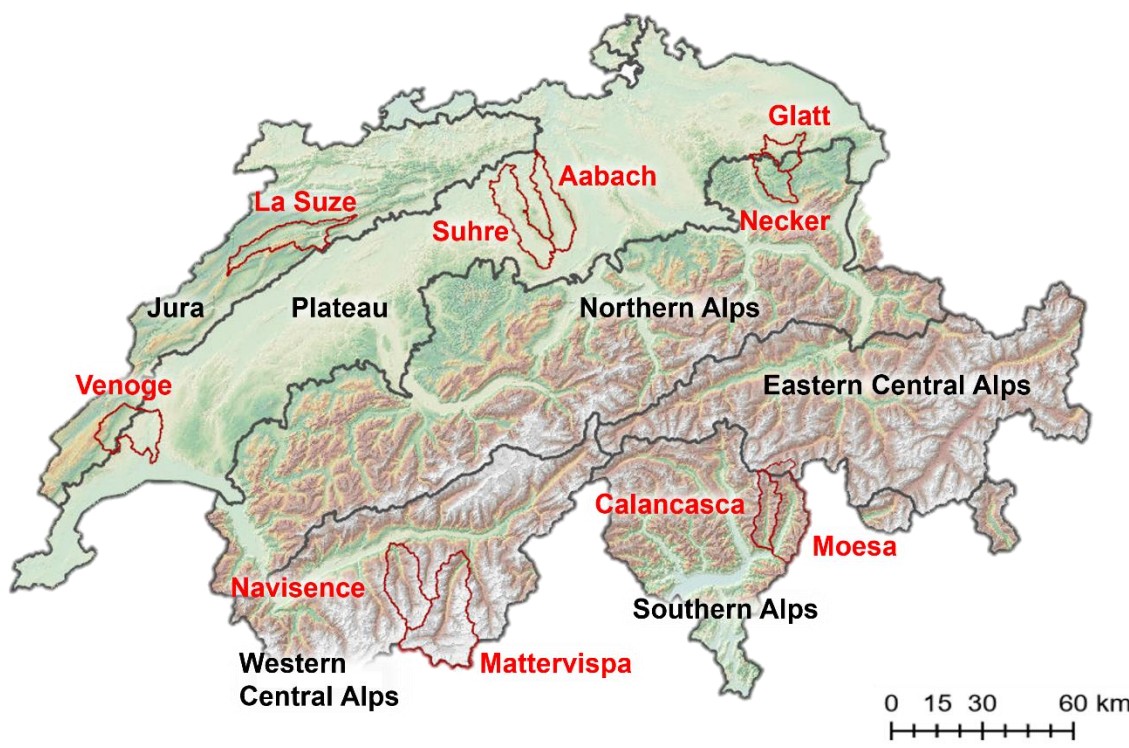

**Figure 2: Location of the ten study catchments in Switzerland. In black: names of the catchments. In red: names of the biogeographical regions.**




**Table 3: Location and physical, geographical and climatic characteristics of the ten catchments selected for this study. For the percentage of historical land use and cover, forested area = F, building area = B, wetland area = W, Lake area = L and glaciers area = G. Please note that the percentages do not sum up to 100% as the rest of the historical land use and cover is represented by fields (i.e., areas that are not included in the other categories). The second to last column reports the results of the plausibility test and indicates the percentage of the historical network situated within the valley bottom. Catchments marked with an asterisk are those represented in the Siegfried map at the 1:50,000 scale. Flow regime type of each catchment is based on the values modelled by the Federal Office of the Environment (FOEN).**

| Biogeographical region | Catchment | Source | Outlet | Historical length of the mainstem (km) | Area (km²) | Δ Elevation catchment and mainstem (m) | % historical land use and cover | % valley bottom over all area | % historical network within valley bottom | Flow regime |
|---|---|---|---|---|---|---|---|---|---|---|
| **Plateau-Northern Alps (Pre-Alpine)** | Necker | Ofenloch | Near Lütisburg, in the Thur River | 31.3 | 125.1 | Catch: 983.4 Main: 737.4 | F: 33.02 B: 0.55 W: 0 L: 0.01 | 14.65 | 50.03% | Nivo-pluvial préalpine |
| | Glatt | Schwellbrunn | Near Niederuzwil, in the Thur River | 24.1 | 90.7 | Catch: 613.4 Main: 446.2 | F: 24.77 B: 0.89 W: 0.81 L: 0.06 | 24.61 | 63.62% | Pluvial supérieur |
| **Plateau** | Aabach | Form Lake Baldeggersee | In Wildegg, in the Aare River | 27.9 | 180.0 | Catch: 531.0 Main: 114.9 | F: 17.80 B: 0.59 W: 0.72 L: 8.71 | 38.09 | 63.23% | Pluvial inferior |
| | Suhre | Near Oberkirch, from Lake Sempach | At the confluence with the Wyna River | 31.3 | 246.9 | Catch: 465.7 Main: 118.0 | F: 23.25 B: 0.53 W: 0.14 L: 11.65 | 39.62 | 79.35% | Pluvial inferior |
| **Plateau-Jura (Jura)** | La Suze | At the top of the Vallon de Saint-Imier | Near Biel, in Lake Biel | 44.5 | 216.3 | Catch: 1179.0 Main: 565.1 | F: 36.47 B: 0.28 W: 0.04 L: 0.01 | 6.70 | 72.48% | Nivo-pluvial jurassien |
| | Venoge | At L'Isle | Between Préverenges and Saint-Sulpice, in Lake Geneva | 37.0 | 231.7 | Catch: 1307.0 Main: 331.9 | F: 32.17 B: 0.29 W: 1.27 L: 0.01 | 17.67 | 77.37% | Nivo-pluvial jurassien |





| | | | | | | | | | | |
|---|---|---|---|---|---|---|---|---|---|---|
| **Southern Alps (Alpine-Mediterranean)** | Moesa* | San Bernardino Pass | Roveredo, at the confluence with Calancasca River | 37.9 | 269.3 | Catch: 2851.3 Main: 1905.9 | F: 31.35 B: 0.36 W: 0.06 L: 0.12 G: 0.85 | 16.42 | 34.49% | Nivo-pluvial meridional |
| | Calancasca* | Southern slope of the Zapporthorn | Roveredo, at the confluence with Moesa River | 30.6 | 141.0 | Catch: 2900.9 Main: 1899.1 | F: 32.61 B: 0.45 W: 0 L: 0.06 G: 1.77 | 9.97 | 25.76% | Nival meridional |
| **Western Central Alps (Alpine)** | Mattervispa* | Confluence of Zmuttbach and Gornera | Stalden, at the confluence with the Saaservispa River | 31.3 (smaller than today's stream – larger glacial cover in the past) | 487.3 | Catch: 3908.9 Main: 1102.4 | F: 44.46 B: 0.13 W: 0 L: 0.04 G: 41.78 | 6.97 | 29.13 | B-glaciaire |
| | Navisence* | Below the Weisshorn | In Chippis, in the Rhone River | 22.8 (smaller than today's stream – larger glacial cover in the past) | 255.5 | Catch: 3981.8 Main: 1296.7 | F: 14.80 B: 0.29 W: 0 L: 0.09 G: 17.64 | 7.77 | 24.66% | A-glacio-nival |




**Table 4: Landscape variables grouped into the three classes shown in Table 1: lithology, topography, and land use/land cover.**

| | Landscape variables | Description | Source |
|---|---|---|---|
| **Lithology** | - Permeable rocks (%) | - Percentage of permeable rock types (e.g., sand, gravel, limestone, dolomite) | Geological map of Switzerland (1:500000) and Hydrogeological map of Switzerland (1:100000) (Swisstopo). |
| | - Impermeable rocks (%) | - Percentage of impermeable rock types (e.g., silt, clay, marble, granite) | |
| **Topography** | - Area (km$^2$) | - Area | Contemporary DEM (digital elevation model, 2 meters resolution; Swisstopo). |
| | - Elevation (m, average) | - Mean elevation | |
| | - Slope (°, average) | - Mean slope | |
| | - Valley area (%) | - Percentage of sub-catchment area occupied by the valley bottom | |
| | - Valley width (m, average) | - Mean width of the valley bottom | |
| **LULC** | - Forested area (%) | - Percentage of sub-catchment area occupied by forest | Siegfried map |
| | - Wetland area (%) | - Percentage of sub-catchment area occupied by wetlands | |
| | - Lake area (%) | - Percentage of sub-catchment area occupied by lakes | |
| | - Building area (%) | - Percentage of sub-catchment area occupied by buildings | |
| | - Glacier area (%) | - Percentage of sub-catchment area occupied by glaciers | |

## 2.4 Statistical analyses

The connectivity metrics and the landscape variables calculated at the small sub-catchments level were used to characterise the historical spatial variability of surface hydrological connectivity between and within the ten study catchments, and its historical drivers. We carried out the statistical analyses keeping the data from the different scales (see paragraph 2.1) separated.



Prior to analysis, the non-parametric Spearman's rank correlation test rho ($\rho$; $\alpha = 0.01$) was applied to test the occurrence of
monotonic relationships among the metrics and landscape variables. When $\rho = \pm 0.90$ only one variable was retained for further analyses. Given the catchments' highly hierarchical structure (Polvi et al., 2020), we accounted for possible spatial autocorrelation by including selected locational variables in the analyses. The names of the catchments help to distinguish them from each other and indicate their specific locations within the territory. The large sub-catchment division reflects the distance of each small sub-catchment from the outlet (i.e., the smaller the ID number of the sub-catchment, the further from
the catchment's outlet). The distance of each sub-catchment from the valley bottom containing the mainstem represented the general "peaks to valley" direction.

To understand if catchments with similar landscape characteristics (e.g., lithology, topography, land use/ land cover) showed similar metrics composition (i.e., similar levels of historical surface hydrological connectivity - objective 1) we used the partial Redundancy Analysis (partial RDA; Borcard et al., 2011). In this analysis, the variability associated with the
locational variables was partialled out from the model (i.e., it was given a fixed value in order to identify correlations between other variables; Borcard et al., 1992). We carried out the analysis employing the ensemble of the connectivity metrics derived for the small sub-catchments as response variables and the set of landscape variables as explanatory variables. after the Prior to the analysis, the metric datasets were transformed with Z-score transformations. To test the significance of the partial RDA axes, we performed a Monte Carlo permutation test (perm = 1000). Variance inflation factor
(VIF) analysis was carried out to check eventual multicollinearity between the predictors (i.e., landscape variables), and variables were removed if VIF $\geq$ 10 (strong multicollinearity; James et al., 2013; O'brien, 2007). We used variance partitioning to quantify the proportion of the total response variability explained by the lithology, topography, LULC, and by the locational variables.

To explore the landscape drivers of the historical spatial patterns of SHC and to investigate possible deviations from the
hypothesized relationships (objective 2), we carried out a series of regression analysis. As for the partial RDA we employed the connectivity metrics derived for the small sub-catchments as response variables and the set of landscape variables as explanatory variables. Multicollinearity between the explanatory variables was checked using variance inflation factor analysis, as described in the previous paragraph. First, we carried out mixed-effect models for each connectivity metric considering the ensemble of the catchments at the 1:25.000 scale, and the ensemble of the catchments at the 1:50.000 scale,
separately. We adopted a nested random effects approach so to check for the influence of the naturally nested structure of the catchments on the values of the different metrics. When the nested effect was not present, we checked the effect of the individual locational variables. We adapted the models to the different metrics statistical distributions (e.g., normal, gamma and Poisson distribution). When the distribution of a metrics showed a large number of zero values, we carried out two-part models (i.e., a first part of the model predicts the probability of a non-zero outcome (e.g., binomial regression), while a
second part models the size of non-zero outcomes (e.g., linear regression)).

We carried out generalized linear model analyses considering the values of each metric for the small sub-catchments within the ten catchments separately. For these analyses, we included the locational variables as fixed factors, so to investigate the



possible internal variability of each catchment. As before, we adapted the models to the different metrics statistical distributions (e.g., normal, gamma and Poisson distribution). When the distribution of a metrics showed a large number of

zero values, we carried out two-part models (see above).

For the statistical analyses we used the R software (R Core Team). In particular, the vegan package (Oksanen et al., 2007) was used to carry out the partial RDA, the lme4 package (Bates et al., 2015) was used to carry out the mixed-effect models and the stats package (R Core Team, 2024) for the generalized linear models. Figures where generated using both ArcGIS Pro and the R software.

## 3. Results

### 3.1 Metrics description for the ten catchments

The catchments in the pre-Alpine region (Necker and Glatt) and those in the Plateau (Aabach and Suhre) presented generally higher total drainage density, with an average of 2.49 ± 0.28 km/km2, compared to the 1.22 ± 0.55 km/km2 of the catchments in the Jura, Alpine-Mediterranean, and Alpine regions (La Suze, Venoge, Moesa, Calancasca, Mattervispa,

Navisence). The same trend is observed for the metrics of shoreline and confluence density. In particular, the Suhre catchment presented a considerably higher confluence density (12.87 ± 21.16 N/km2) compared to the other catchments (1.94 ± 1.19 N/km2). The highest stream order at the outlet was observed for the Glatt catchment, as a 6th order stream, while most of the other catchments were 5th order streams. The proportion of 1st order streams ranged from 61.5% for the Calancasca catchment to 40.4% for the Suze catchment. The Suze catchment and the catchments in the Plateau and the

Alpine-Mediterranean region presented a lower percentage of shoreline bordering forested area (17.1 ± 5.5 %) compared to the Venoge catchment and the catchments in the pre-Alpine and Alpine regions (45.9 ± 13.1 %). The catchments in the Plateau region, together with the Glatt and Suze catchments presented a higher percentage of shoreline bordering buildings (5.8 ± 1.5 %) compared to the catchments in the Alpine regions and the Necker and Venoge catchments (1.5 ± 0.7 %). The Venoge catchment was the one with the highest percentage of shoreline bordering wetlands (8.7%) followed by the Aabach

(1.7%) and Suze catchments (1.3%). The spatial distribution of the different metrics of connectivity within each catchment is shown in Fig. S1 in supplemental material.

### 3.2 Metrics heterogeneity at sub-catchments level – Partial RDA

The Spearman correlation analyses among the connectivity metrics for the ensemble of the catchments at both the 1:25000 and at the 1:50000 scale revealed high positive monotonic relationship between drainage density and shoreline density (ρ =

1.00), and a high negative monotonic relationship between the percentages of the shoreline bordering forested areas and fields, respectively (ρ = -0.95). For this reason, only one of the two correlated pairs of metrics were retained for the partial RDA and the mixed-effect models. Here, we retained drainage density and the percentage of shoreline bordering forested





areas. The Spearman correlation analyses for the explanatory landscape variables did not show monotonic relationships higher than $\rho = + 0.90$ or lower than $\rho = - 0.90$. No explanatory variables were excluded after the VIF analysis.

Figure 3 shows the results of the partial RDAs and Venn's diagram for the catchments at the 1:25000 scale (Fig. 3 a and b) and at the 1:50000 scale (Fig. 3 c and d). For the catchments at the 1:25000 scale, the total sub-catchments variation explained by the partial RDA is 23.5%. 19.5% of the variation is conditioned by the spatial variability. The RDA ordination is significant in predicting all the variables (P < 0.001). The first RDA axis covers 12.9 % of the explained variation, while the second RDA axis covers 7.4 %. Both axes result as significant (P < 0.001) after the permutation test. For the catchments

at the 1:50000 scale, the total sub-catchments variation explained by the partial RDA is 28.2%. 19.9% of the variation is conditioned by the spatial variability. The RDA ordination is significant in predicting all the variables (P < 0.001). The first RDA axis covers 16.8 % of the explained variation, while the second RDA axis covers the 9.0 %. Both axes result as significant (P < 0.001) after the permutation test. It is worth to note that the third RDA axis (not shown) for both analyses also carried a considerable proportion of the explained variation (4.8% for the sub-catchments at the 1:25000 scale and 5.2%

for the sub-catchments at the 1:50000 scale).

After accounting for the effect of the locational variables, the majority of the sub-catchments at both the 1:25000 scale and the 1:50000 scale clustered together around the 2D ordination origin in their respective analyses. The sub-catchments in Venoge are the only ones to show a mild clustered behavior along the second axis. For a clearer analysis of the sub-catchments at a 1:50000 scale, we removed the only two sub-catchments with shoreline bordering wetlands (from the Moesa

catchment). This improves the resolution of the (dis)similarities among the remaining sub-catchments. For both the sub-catchments at the 1:25000 scale and at the 1:50000, the LULC variables drive most of the difference observed among the metrics composition of the sub-catchments. For the sub-catchments at the 1:25000 scale, the LULC variables explain 13.4% of the variability (see Venn diagram in Fig. 3 b), while for the sub-catchments at the 1:50000 scale, the LULC variables explain 10.9% of the variability alone and 12.1 % of the dataset's variability jointly to the locational and topographic

variables (see Venn diagram in Fig. 3 d). More in details, for the sub-catchments at the 1:25000 scale, the percentage of area covered by wetlands is the main variable (among the LULC variables and in general) associated with differences in the sub-catchments' metrics composition (longer arrow, along the second axis). The occurrence of sub-catchments with particularly high percentage of wetland areas, mainly belonging to the catchments in the Plateau (Aabach and Suhre), and to the Venoge catchment, is particularly visible in the bi-plot ordination (Fig. 3 a). The percentage area covered by buildings is also an

important variable (mostly along the first axis), driving the sub-catchments variability mostly within the catchments in the pre-Alpine region (Necker and Glatt) and the Plateau (Aabach and Suhre). Sub-catchments associated to a high percentage of building area seem to be also associated to a higher percentage of permeable rock. For the sub-catchments at the 1:50000 scale the percentage of area covered by forest is the main variable (among the LULC variables and in general) associated with differences in the sub-catchments' metrics composition (longer arrow, along the second axis). The sub-catchments

variability of all catchments is particularly driven by this variable and, to a lesser extent, by the percentage of area covered by buildings (particularly visible for the sub-catchments in Moesa and Calancasca, in the Alpine-Mediterranean region). The



topography explains 4.9% and the 7.9% of the variability for the sub-catchments at the 1:25000 scale and at the 1:50000 scale, respectively (see Venn diagrams in Fig. 3 b and d). For the catchments at the 1:25000 scale, the percentage of valley area and the valley width are associated to sub-catchments variability mostly along the third axis (not shown) and, in

particular, drive the variability within the Suhre and Glatt catchments. For the sub-catchments at the 1:50000 scale, the percentage of valley area and the valley width are associated to sub-catchments variability mostly along the first axis and, in particular, drive the variability within the Alpine-Mediterranean catchments (Moesa and Calancasca). For the sub-catchments at the 1:25000 scale, mean elevation and slope play a synergic role in driving sub-catchments' variability along the first axis, while for the sub-catchments at the 1:50000 scale the average elevation drives the variability mostly along the

first axis and the average slope mostly along the second axis. It is interesting to note that, for the catchments at the 1:50000 scale, the catchments at a lower elevation are also associated with higher percentages of both permeable and impermeable rocks.



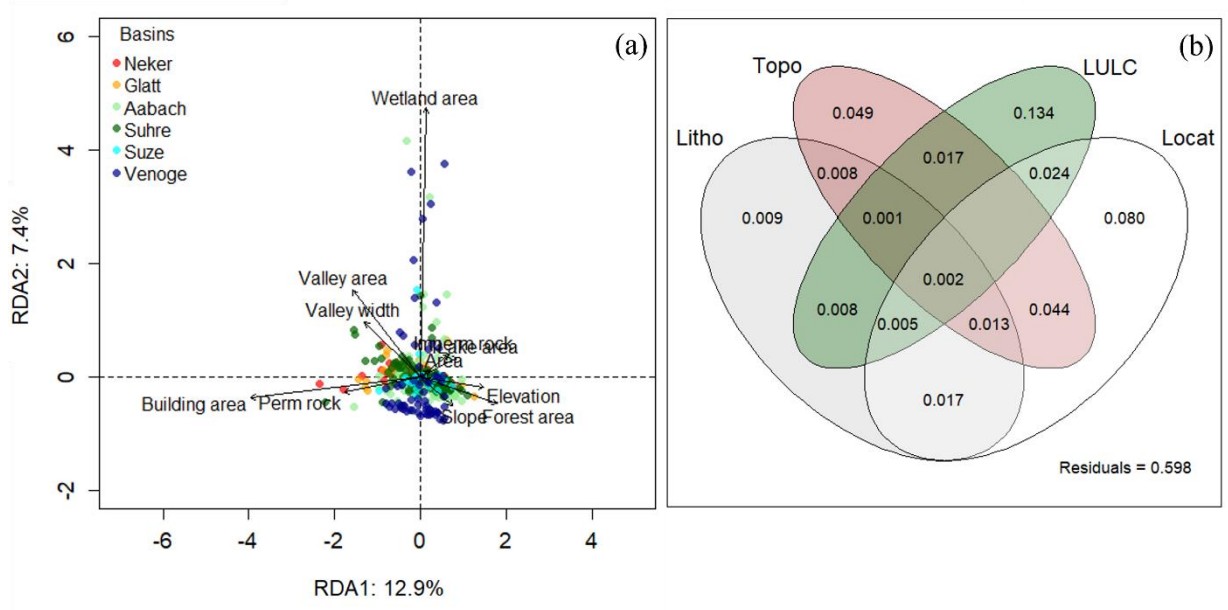

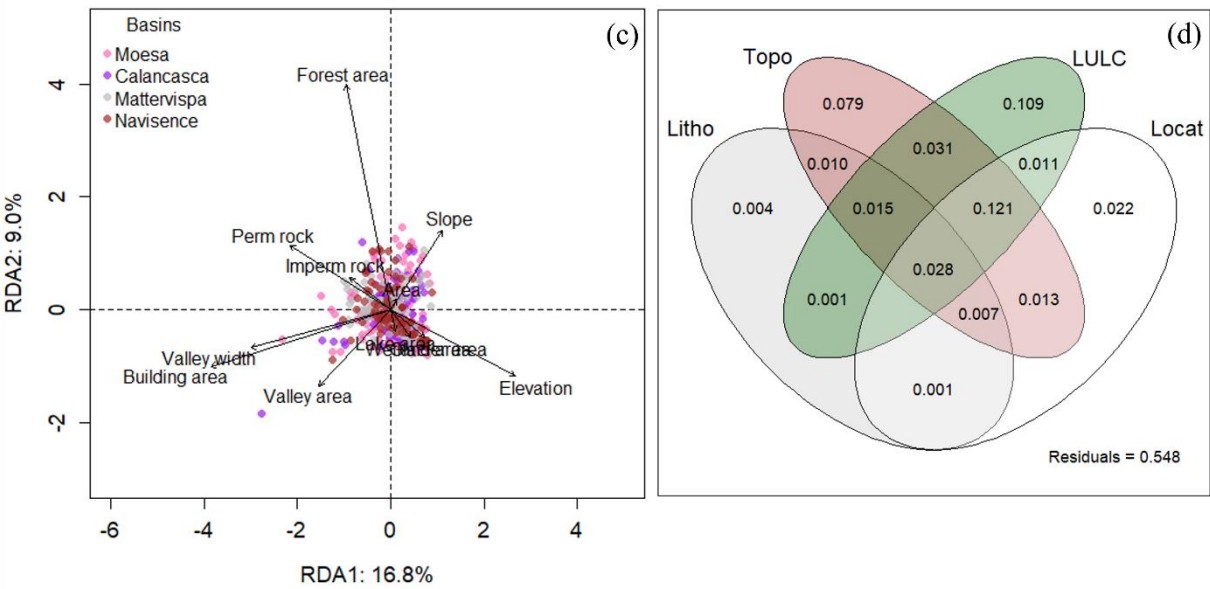

**Figure 3: Partial RDAs bi-plots showing the relationship between sub-catchments' connectivity metric composition and landscape variables for the catchments at the 1:25000 scale (a) and for the 1:50000 scale (b). Cond: variability conditioned by the locational variables; Constr: variability constrained by the landscape variables. The percentages indicated close to the axes labels indicate the percentage of constrained variability explained by the axes. (b) and (d): Venn's diagrams showing the results of the variance partitioning analyses for the catchments at the 1:25000 scale and for the 1:50000 scale, respectively. The values refer to the proportion of the total response variability explained by the different landscape variables (Litho, Topo, LULC) and by the**




**locational variables (Locat). Values in the shared areas indicate the proportion of explained variability shared among two or more factors.**

### 3.3 Landscape drivers of the historical patterns of surface hydrological connectivity - Regression analyses

### 3.3.1 Mixed-effect models

The results of the mixed-effect models for each connectivity metric considering the ensemble of the catchments at the
1:25.000 and the 1:50000 scales are shown in Table 5 a and b, respectively.

The metrics' variability explained by the locational variables, and in particular by the hierarchical structure of the catchments, is generally higher for the catchments at the 1:25000 scale. For these catchments, most of the metrics' variability is explained by the sub-catchment location within the catchment in term of distance from the outlet, and by the variability among the different catchments. This is particularly true for the drainage density, confluence density, the percentage of
shoreline bordering forest and the non-zero portion of the percentage of shoreline bordering wetlands. For the catchments at the 1:50000 scale, the locational variables explain a high percentage of the metrics variability in the case of confluence density and for the percentage of shoreline bordering forest. The variability in these two metrics is particularly explained by the sub-catchment location within the catchments in terms of their distance from the valley.

Rock lithological characteristics (high and low permeability) are significantly related to several metrics, especially for the
sub-catchments at the 1:25000 scale. We found an unexpected negative relationship between the percentage of impermeable rock and the non-zero portion of the percentage of the shoreline bordering wetlands (Table 5 a). We also found unexpected positive relationship between the percentage of permeable rock and the mode of the stream order for both the sub-catchments at the 1:25000 and 1:50000 scale. The relationships of the topographic variables with the different metrics mostly correspond to our hypothesized relationships, with the only opposite relationships being an observed negative relationship between the
percentage of area covered by the valley bottom and the non-zero portion of the percentage of shoreline bordering wetlands for the 1:25000 scale sub-catchments and with the mode of the stream order for the 1:50000 scale sub-catchments, and observed positive relationships between the average slope and the maximum stream order and the presence/ absence of shoreline bordering buildings for the 1:25000 scale sub-catchments, and a positive relationship between the average valley width and confluence density for the 1:25000 scale sub-catchments. Finally, the LULC variables show some deviations from
our hypothesized relationships with the connectivity metrics. In particular, the relationship between the percentage of sub-catchment area covered by building and drainage density, confluence density, and the metrics related to stream order was always positive for both the sub-catchments at the 1:25000 scale and at the 1:50000 scale. For the sub-catchments at the 1:25000 scale, the percentage of sub-catchment area covered by forest show positive relationships with the non-zero portion of the percentage of shoreline bordering buildings.




**Table 5: Outputs of the mixed-effect models for the catchments at the (a) 1:25000 scale and (b) 1:50000 scale. The cells colored in light gray show significant positive relationships, while those colored in dark gray show significant negative relationships. When two columns are present below a single metric, it indicates that a two-part model has been carried out, and results are shown for the test on the "presence-absence" of a certain metric (P/A; binomial regression), and for the "non-zero" portion of that metric (non-zero; linear regression). The diagonal bar indicates that the observed relationship is contrary to our hypothesised relationship (see Table 1). The first seven rows show the proportion of the data variability (expressed in % of the total variability) explained by the locational variables. The last two rows report the explanatory power of the models expressed in conditional and marginal $R^2$. Conditional $R^2$ refers to the explanatory power of the model when considering both random and fixed effects, while marginal $R^2$ refers to the explanatory power of the model when considering only the fixed effects. Please note that the variables "Shoreline wetlands" and "Wetland area" have been removed from the analyses for the catchments at the 1:50000 scale as wetlands were found to be only present in two sub-catchments of the Moesa catchment.**

| *(a)* | Drainage density | Confl. density | Stream order (max) | Stream order (mode) | Shoreline forest | Shoreline wetland | | Shoreline buildings | |
|---|---|---|---|---|---|---|---|---|---|
| *Dist valley:dist outlet:basin* | | | | | 1.2 | | | | |
| *Dist valley: basin* | | | ~0.0 | 0.7 | | | | 3.1 | |
| *Dist outlet: basin* | 16.5 | | | | 6.6 | | 5.3 | | |
| *Dist outlet* | | | | | | | | 2.7 | |
| *Basin* | 6.7 | 26.7 | 0.3 | 2.3 | 43.8 | 6.0 | 12.0 | 3.4 | 4.8 |
| | | | | | | P/A | Non-zero | P/A | Non-zero |
| *Perm rock* | 0.03 | -0.18 | 0.06 | 0.16 ** | -3.54 * | -0.19 | -0.71 *** | -0.12 | 0.27 *** |
| *Imperm rock* | -0.11 | -0.22 | -0.03 | 0.03 | 0.43 | 0.72 | -0.44 * | -0.27 | 0.35 ** |
| *Area* | -0.15 * | -0.05 | 0.06 | -0.05 | -3.16 * | 0.82 *** | 0.10 | 1.26 *** | -0.01 |
| *Elevation* | 0.01 | -0.17 | -0.20 *** | -0.28 *** | 8.43 *** | -1.00 * | -0.23 | -0.93 *** | -0.13 |
| *Slope* | 0.33 ** | 0.27 * | 0.14 * | 0.10 | -4.66 | -0.59 | 0.12 | 0.58 * | 0.26 |
| *Valley area* | 0.79 *** | 0.07 | 0.08 | -0.03 | 0.55 | 0.22 | -1.05 *** | 0.19 | -0.09 |
| *Valley width* | -0.12 * | 0.21 ** | -0.01 | 0.01 | -0.82 | 0.00 | 0.57 ** | -0.30 | 0.05 |



| | | | | | | | | | |
|---|---|---|---|---|---|---|---|---|---|
| Forest area | -0.15 * | 0.14 | -0.03 | -0.03 | 9.11 *** | -0.07 | 0.13 | -0.15 | 0.31 *** |
| Building area | 0.10 | 0.20 ** | 0.04 | 0.11 ** | -3.64 ** | -0.45 | -0.17 | 1.57 *** | 0.74 *** |
| Lake area | -0.13 * | -0.20 ** | -0.10 ** | -0.02 | 0.01 | 0.03 | -0.06 | -0.30 * | 0.10 |
| Wetland area | -0.7 | 0.11 | 0.02 | 0.04 | -2.85 ** | 5.61 *** | 0.47 *** | -0.15 | -0.01 |
| Conditional $R^2$ | 0.45 | 0.36 | 0.15 | 0.25 | 0.61 | 0.93 | 0.74 | 0.56 | 0.47 |
| Marginal $R^2$ | 0.29 | 0.13 | 0.13 | 0.16 | 0.19 | 0.92 | 0.69 | 0.53 | 0.43 |

Statistical significance of regression coefficients: ***p value < .001; **p value < .01; *p value < .05; The ":" notation is used to indicate the locational variables when nested.

| (b) | Drainage density | Confl. density | Stream order (max) | Stream order (mode) | Shoreline forest | Shoreline buildings |
|---|---|---|---|---|---|---|
| Dist valley:dist outlet:basin | | | | | | 0.7 |
| Dist valley:dist outlet | | | | 0.4 | | |
| Dist valley:basin | | 69.8 | | | | |
| Dist outlet:basin | | | 0.1 | | | ~0.0 |
| Dist valley | | | | | 21.5 | |
| Dist outlet | | | | 0.8 | | |
| Basin | 0.65 | ~0.0 | 0.1 | | | 5.2 |

| | Drainage density | Confl. density | Stream order (max) | Stream order (mode) | Shoreline forest P/A | Shoreline buildings Non-zero |
|---|---|---|---|---|---|---|
| Perm rock | -0.06 | -0.05 | 0.03 | 0.08 * | 0.07 | 0.34 * | 0.16 |
| Imperm rock | 0.04 | 0.06 | 0.03 | -0.03 | 0.10 | -0.17 | 0.08 |
| Area | -0.03 | -0.07 | 0.05 | -0.04 | -0.26 ** | - | 0.35 |





| | | | | | | | |
|---|---|---|---|---|---|---|---|
| *Elevation* | -0.14 | -0.03 | -0.24 ** | -0.37 *** | 0.24 | 0.28 | 0.15 |
| *Slope* | 0.20 *** | 0.18 | 0.02 | -0.08 | 0.27 | -0.00 | 0.01 |
| *Valley area* | 0.40 *** | 0.40 ** | -0.02 | -0.14 ** | 0.06 | -0.22 | 0.10 |
| *Valley width* | -0.10 * | 0.07 | 0.01 | 0.09 * | -0.04 | -0.09 | 0.10 |
| *Forest area* | 0.02 | 0.17 | 0.02 | -0.02 | 1.05 *** | 0.32 | -0.06 |
| *Building area* | 0.09 | 0.22 * | 0.05 | 0.02 | -0.08 | 1.79 *** | 0.46 *** |
| *Lake area* | -0.03 | -0.03 | -0.01 | -0.02 | -0.25 | -0.84 | -0.25 |
| *Glacier area* | -0.16 *** | -0.22 | -0.05 | 0.06 | -1.02 *** | -1.23 | -2.54 ** |
| *Conditional R²* | 0.41 | 0.72 | 0.27 | 0.25 | 0.54 | NA | 0.41 |
| *Marginal R²* | 0.41 | 0.15 | 0.26 | 0.22 | 0.42 | 0.32 | 0.38 |

Statistical significance of regression coefficients: ***p value < .001; **p value < .01; *p value < .05;

The ":" notation is used to indicate the locational variables when nested.

### 3.3.2 Generalized linear models

The results of the generalized linear models for the derived connectivity metrics of the different catchments are shown in Fig. 4. Spearman correlation analyses among the connectivity metrics within each catchment showed high positive monotonic relationship between drainage density and shoreline density ($\rho = 1.00$). A high negative monotonic relationship between the percentages of the shoreline bordering forested areas and fields was found for most catchments ($\rho \leq -0.95$), except for the Aabach, Suze, Venoge and Mattervispa catchments. The Spearman correlation analyses for the explanatory landscape variables within each catchment did not show monotonic relationships higher than $\rho = +0.90$ or lower than $\rho = -0.90$. Depending on the analyzed catchment, VIF analyses identified different factors with high values, indicating varying sources of multicollinearity across the datasets. In particular, the main variables that have been omitted are sub-catchment distance from the outlet for Necker, Glatt and Suhre catchments, and the average elevation for the Venoge and Navisence catchments.

The locational variables (for this analysis, sub-catchments' distance from the outlet and from the valley) are not always among the significant explanatory variables for the metrics' variability within all the considered catchments, and result to be





significant mostly for the catchments in the Plateau (Aabach and Suhre), the Venoge catchment and for the catchments in the Alpine areas in general (Moesa, Calancasca, Mattervispa, Navisence). The locational variables result as significant

explanatory variables mostly for drainage and confluence density, and the metrics relative to stream order (maximum and mode order values). Unexpected positive relationships between the sub-catchments' distance from the outlet and the maximum and mode of the stream order are observed for the catchments in the Alpine-Mediterranean region (Moesa and Calancasca) and for the Mattervispa catchment.

Rock lithological characteristics (high and low permeability) show diverse relationships with the different metrics. High

percentage of permeable rocks is found to be highly positively related to the stream order mode in the Glatt catchment. The percentage of impermeable rocks showed an unexpected negative relationship with drainage density and confluence density for the Necker and Glatt catchments, respectively. The relationships between the topographic characteristics of the sub-catchments and their metrics' distribution largely align with our hypotheses. The area of the sub-catchments is always positively related to most of the metrics (except for the mode of the stream order in the Suhre catchment), and in particular to

confluence density for the catchments in the Alpine region (Mattervispa and Navisence), the maximum stream order for the catchments in the Plateau (Necker and Glatt) and for the Moesa catchment, and to the percentage of shoreline bordering buildings for the Pre-Alpine catchments (Necker and Aabach), the Suhre catchment, and the Alpine catchments (Mattervispa and Navisence), where it explains the presence/ absence of this metric. A higher percentage of the area covered by the valley bottom is positively related to all catchments except those in the Jura (Suze and Venoge), and to confluence density for the

catchments in the Plateau (Aabach and Suhre), and the Venoge and Moesa catchments. For the Necker catchment, the percentage of the area covered by the valley bottom is found to be negatively related to the confluence density. The average valley width is related to the different metrics of longitudinal connectivity with mixed relationship sign depending on the analyzed catchment. As expected, the average elevation is always negatively related to both stream other metrics in the catchments in the pre-Alpine region (Necker and Glatt) and both Alpine regions (Calancasca, Mattervispa and Navisence).

However, average elevation shows unexpected negative relationship with drainage and confluence density for Venoge and Moesa catchments, respectively. The average slope also shows some unexpected relationships, and in particular a positive relationship with the maximum stream order, and a negative relationship with confluence density for the Necker catchment. The LULC variables relationship with the different metrics within the different sub-catchments is diverse. However, the percentage of area covered by glaciers (mostly present in the catchments in the Alpine regions) is always negatively related

to drainage and confluence density, and to the percentage of shoreline bordering forest, as hypothesized. The percentage of area covered by forest is mainly positively related to the percentage of shoreline bordering forest, and in particular for the catchments in the pre-Alpine region (Necker and Glatt) and both Alpine regions (Moesa, Calancasca, Mattervispa and Navisence). However, a negative relationship is found for the Aabach catchment (against our hypothesized relationship). Moreover, the percentage of area covered by forest is found to be positively related to the percentage of shoreline bordering

buildings and fields for the Aabach catchment, which is also against our hypothesized relationships. The percentage of area covered by buildings is positively related to the percentage of shoreline bordering buildings for all catchments except





Venoge and Mattervispa. However, it is also found to be positively related to drainage and confluence density, maximum and mode of the stream order for a variety of catchments (against our hypothesized relationship). The percentage of area covered by lakes and wetlands presented quite diverse relationships depending on both the considered metric and catchment.











**Figure 4: Results of the generalized linear model analyses for the ten considered catchments. We indicate as "positive" and "negative" the relationships with a statistical significance of the coefficients up to *p* value < .05, and as "strong positive" and "strong negative" the relationships with a statistical significance of the coefficients up to *p* value < .001. When the size of the tile is half, it indicates that two-part modelling was used. The relationships found for the binomial part of a two-part modelling ("presence-absence" of a certain metric) are indicated with a black dot, while half-tiles without the black dot report the relationships found for the non-zero portion of the model (linear regression).**

## 4. Discussion

We investigated the similarity and dissimilarity among the ten catchments based on their metrics composition and landscape characteristics through partial RDA. For the sub-catchments at both scales (1:25000 and 1:50000, respectively), we identified the landscape characteristics that drove most of their variability. We investigated these relationships further by carrying out a series of regression analyses with different levels of data aggregation. Here, we discuss the main patterns and relationships we observed, in relation to the hypotheses exposed in Table 1.

### 4.1 Longitudinal hydrological connectivity

For the catchments at the 1:50000 scale, the patterns of longitudinal connectivity were found to be mostly driven by the topography (i.e., average slope). While we found strong positive relationships between the percentage of area occupied by the valley bottom and the drainage and confluence density, we also found negative relationships between the average width of the valley bottom and drainage density (Table 5 b and Fig. 4). These relationships were of the opposite direction when relating these two valley bottom characteristics with the mode of the stream order (Table 5 b). These results may reflect the heterogeneity in valley morphology that characterises these catchments, with numerous narrow V-shaped valleys at higher elevation, and wider valley bottoms at lower elevation, but also where high mountain plateaus are present. The valley bottom morphology (e.g. confinement) and distribution have been shown to be among the primary drivers of river morphology and behaviour (Fryirs et al., 2016; O'brien et al., 2019). This could also explain the positive relationships between the sub-catchments' distance from the outlet and the mode of the stream order for the catchments in the Alpine-Mediterranean region (Moesa and, Calancasca) and for the Mattervispa catchment (Fig. 4). In these catchments, the presence of numerous low-order connected channels at high elevation favours the formation of high stream orders relatively far from the catchment's outlet. Moreover, in the Mattervispa catchment we noted the presence of particular irrigation channels built to transport water along the side of the valleys, called "bisse" (Reynard, 1997). As we assigned the first stream order to these channels, this may have been further contributed to the observed positive relationships between the stream order metrics and the sub-catchments' distance from the outlet in this catchment. The characteristics of the valley bottom were also among the main drivers of longitudinal connectivity for the catchments at the 1:25000 scale (Table 5 a and Fig. 4). However, for these catchments, we also found that other landscape characteristics played a considerable role in driving the patterns of longitudinal connectivity. Both drainage and confluence density, and the maximum stream order were found to be negatively correlated with the percentage of area covered by lakes. The presence and location of lakes within a catchment create natural



discontinuities, leading to significant longitudinal changes of the physical and ecological characteristics of rivers (Jones, 2010; Seekell et al., 2022). An unexpected highly significant positive relationship was observed between the percentage of the area covered by buildings and confluence density (Table 5 a). This relationship could also be observed for specific catchments in the Plateau region, such as Aabach (Fig. 4) and Suhre (Fig. 3 a). Within those catchments, the sub-catchments with a higher percentage of the area covered by buildings were also found to be also associated with a high percentage of area covered by the valley bottom and a large valley width (Fig. 3 a). The sub-catchments showing this combination of features were characterized by an extensive network of man-made ditches, and mostly located in the lower part of the two catchments. The areas characterized by these ditches were called "*Wässermatten*" and were part of a traditional irrigation system (Federal Office for Culture; Foc, 2024). The *Wässermatten* were irrigated multiple times a year, with the water also carrying manure to fertilize the fields. Maintenance of the main ditches was a collective responsibility of the cooperatives, while private individuals took care of the smaller side ditches (Federal Office for Culture; Foc, 2024). As for the case here described, the positive relationship between the percentage of the area covered by buildings and longitudinal connectivity could reflect the presence of areas that had been already modified by anthropogenic activities. However, the same could be said in case of a negative relationship would have occurred, mostly depending on the type of human activity and exploitation (Weber et al., 2009). Finally, for both the catchments at the 1:25000 scale and 1:50000 scale, we found an unexpected positive relationship between the percentage of permeable rock and the mode of the stream order (Table 5 a and b). In all studied catchments, the majority of the sub-catchments characterized by a high percentage of permeable rock were the ones closer to the mainstem. This may suggest that the occurrence of permeable rocks does not prevent the development of high stream orders when permeable rocks are mainly present in the downstream valleys.

**4.2 Lateral hydrological connectivity**

Within the catchments at the 1:50000 scale, wetlands were only present in two sub-catchments within the Moesa catchment, located close to the source area. For the catchments at the 1:25000 scale, the percentage of shoreline bordering wetlands was positively driven by the presence of the wetlands within the sub-catchments and by the average valley width (Table 5 a). The percentage of area covered by wetland was found to be an important factor driving the sub-catchments' variability in the Plateau and Jura regions (Fig. 3 a). These two regions were historically characterised by a large amount of wetlands (Gimmi et al., 2011; Stuber and Bürgi, 2019). A positive relationship between the percentage of area covered by wetlands and the percentage of the shoreline bordering them was found in particular for the Aabach and Venoge catchments (Fig. 4). In Aabach, the largest wetlands were mostly located on the lakeshore, whereas in the Venoge they were found at the foothill of the Jura Mountains, although the presence of wetlands was quite widespread in this catchment. Despite the Glatt catchment was the one with the second higher percentage of wetland area (mostly located within the lower part of the catchment, at the start of the Plateau region), we did not detect the expected relationship between the percentage of area covered by wetlands and the percentage of the shoreline bordering them within this catchment (Fig. 4). Differently from the other catchments, in



Glatt we found a negative relationship between the percentage of area covered by wetlands and a metric of longitudinal connectivity such as drainage density. This could be related to the fact that, unlike the wetlands in the Aabach and Venoge catchments, the wetlands in the Glatt catchments were mostly peatlands as it can be seen by the symbols used to depict them in the map. Their historical exploitation for peat extraction, only banned since 1987, required the drainage of the peatlands,

with possible repercussion on the surrounding area (Charman, 2009; Wüst-Galley et al., 2019). The observed negative relationship between the percentage of the area covered by the valley bottom and the percentage of shoreline bordering wetland as well as the negative relationship between the percentage of impermeable rocks and the percentage of shoreline bordering wetland are of difficult interpretation (Table 5 a). For all the catchments at the 1:50000 scale, the percentage of shoreline bordering forest was positively related with the percentage of area covered by forest (Table 5 b). This variable

drove a considerable part of the variability of the catchments in the Alpine regions, with the sub-catchments with the higher percentage of forested area being those at the lower elevation (Fig. 3 c). The same positive relationship between the percentage of shoreline bordering forest and the percentage of area covered by forest was found for the catchments at the 1:25000 scale (Table 5 a), and in particular in the pre-Alpine region (Fig. 4). However, for the catchments at the 1:25000 scale, the percentage of shoreline bordering forest was also positively related to the average elevation (Table 5 a). This

reflects the natural distribution of forest stands in catchments with different physiographic characteristics (Klinge et al., 2015). The presence and location of forested shorelines have been shown to play a major role in reducing local and downstream water temperature, with repercussion on species behavior and movement patterns (Roon et al., 2021). It is interesting to note the negative relationship between the percentage of shoreline bordering forest and average elevation (for the Suhre catchment) and between the percentage of shoreline bordering forest and the percentage of forested area (for the

Aabach catchment). Moreover, for the Aabach catchment, the percentage of shoreline bordering fields showed a strong positive relationship with the percentage of forested area. These unexpected relationships can be due to the fact that, within these two catchments, the majority of the of the forested area is situated in the lower part of the catchment, but the immediate surrounding of the stream shoreline is mostly represented by fields, and in particular by the overmentioned wässermatten. Because of the lateral flooding resulting from the irrigation system in the wässermatten, the lateral connectivity within these

areas could still be considerable (Zhao et al., 2024). Finally, for the catchments at both scales, the percentage of shoreline bordering building was found to be mainly related to the percentage of area covered by buildings (Table 5 a and b). This was observed for most catchments (Fig. 4), and possibly reflecting the necessity to build settlements close to a water source (Fang et al., 2018).

**5.3 The role of locational variables**

As expected, and corroborated by the results of the RDA analysis (Fig. 3), the locational variables played a considerable role in defining the patterns of connectivity among the studied catchments (see conditioned variability in Fig. 3 a and c) (Larsen et al., 2021). The catchments at the 1:25000 scale were quite heterogeneous, with marked physiographic differences when compared to each other (Table 3). These differences played a role in explaining the variability of the different connectivity





metrics, in particular when considering the confluence density, the percentage of shoreline bordering forest and bordering wetlands (Table 5 a). On the contrary, the catchments at the 1:50000 scale were more homogeneous in their physiographic characteristics (Table 3). For these catchments, the variability of the different connectivity metrics was mostly explained by the internal distribution of the sub-catchments and in particular by the distance from the valley (Table 5 b, in particular for confluence density and percentage of shoreline bordering forest). This reflects the primary role played by the topography in controlling large-scale hydrological processes in the Alpine catchments (Jencso et al., 2009). When considering the driver-metric relationships within the single catchments (Fig. 4), the locational variables did not seem to play a constant role in explaining the metrics variability, but rather be catchment specific. In some cases, eventual redundancy between the locational variables and the landscape variables has been controlled by carrying out VIF analysis and excluding the locational variable from the regression analyses (see paragraph 3.3.2).

We tested the plausibility of the hydro-morphological metrics for a given sub-catchment by calculating to which proportion the hydrographic network depicted on the Siegfried maps aligns with the valley bottom calculated from the contemporary DEM (Table 3). It can be seen how this value varies for the different catchments, with those in the Alpine regions presenting the lower percentages of network situated within the valley bottom (between ~25 and ~35 %). This is probably due to the fact that, because of their scarce accessibility, areas at higher elevation and with a narrow valley bottom were intrinsically more difficult to map (Sampaio and Rocha, 2022; Pena et al., 2018). In fact, even within the Alpine catchments, the areas at higher elevation behaved systematically worse than the areas at lower elevations where a wider valley bottom is present. Despite this, it is important to note that the calculated low proportions of the hydrographic network aligning with the valley bottom where mostly due to positional errors, where the entire network is uniformly misaligned, rather than topological errors, which would involve incorrect or disrupted relationships between the elements of the network and with the surrounding landscape (Leyk et al., 2005).

## 5. Conclusions

In this contribution, we have explored the historical spatial patterns of surface hydrological connectivity in catchments with different physiographic characteristics. To do so, we derived a series of hydromorphological metrics and used them as proxies for large-scale longitudinal and lateral connectivity. We have characterised the historical landscape drivers of connectivity and discussed the results in relation to hypothesised driver-metrics relationships. Most of the observed connectivity patterns could be explained by landscape drivers in a manner that aligned with our hypothesized connectivity-driver relationships. However, even though the landscapes and stream networks within the ten catchments had not yet been severely altered by extensive urbanization or extreme fragmentation from dam construction in the late 19th century, human activities like historical irrigation practices and peat digging already influenced the historical patterns of surface hydrological connectivity. These observations are valuable for understanding the history of human exploitation of the territory and its environmental impact and legacy.




We showed how, with due consideration of the inherent limitations of historical maps, the information they provide could enhance our understanding of large-scale processes. Historical maps offer a snapshot of past landscapes and hydrological systems, upon which today's landscapes and rivers have developed (Turner, 2010). The multi-metric approach has proven fundamental for the investigation of hydrological processes at large scale, which are inherently difficult to measure direcly

(Heasley et al., 2019; Palmer and Febria, 2012; Torgersen et al., 2022). To guarantee the healthy functioning of the entire hydrological system, it is crucial to preserve important processes such as hydrological connectivity on a large scale (Abell et al., 2007). This is fundamental for achieving successful restoration measures at the reach scale, which are the most commonly implemented. These measures are more likely to fail if processes and dynamics at the catchment scale are not considered, especially when the catchment scale aligns with the scale of degradation (Kail et al., 2015; Miller et al., 2010;

Rachelly et al., 2021).

## Data availability

The data that support the findings of this study are available from the corresponding author upon reasonable request.

## Author contribution

Marta Antonelli designed the study, performed research, analysed data, wrote the paper. Victoria Scherelis analysed data and contributed new methods. Christine Weber conceived and designed the study.

## Competing interests

The authors declare that they have no conflict of interest.

## Acknowledgements

We would like to thank Tara Behnsen for her assistance in correcting the GIS data, Rosi Siber for her support with GIS data analysis, and Andreas Scheidegger for his suggestions and guidance on the statistical approach.

## Financial support

This work was funded by the Swiss National Science Foundation (SNSF) (No. 188692).



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
