# Peer review of "Metric-based analysis of the historical drivers of surface hydrological connectivity"

_EGUsphere, 2025_

## Author Comment (AC1)

**We thank the Reviewer for highlighting these crucial points.**

**Please find our detailed responses below.**

**Reviewer #1:**

This article presents a novel and ambitious analysis of historical surface hydrological connectivity across ten Swiss catchments, using digitized versions of the Siegfried map developed between 1870 and 1926. The authors derive a set of hydromorphological metrics as proxies for surface hydrological connectivity, including drainage density, confluence density, stream order metrics, and shoreline metrics related to different land cover types. These metrics are then statistically related to landscape variables such as lithology, topography, and historical land use/land cover to identify the main drivers of hydrological connectivity in the late 19th century. The study finds that topographic features, particularly valley morphology and slope, are consistent primary drivers of longitudinal connectivity, while wetlands and forested areas play key roles in lateral connectivity. Several unexpected relationships are identified, such as the positive association between building area and hydrological metrics in some catchments, which are interpreted in the context of historical irrigation and land use practices.

I have following comments for improvement before possible publication.

Major Comments

1) The methodology is well developed and generally appropriate for the research questions posed, but there are areas where improvements could increase scientific robustness and interpretive power. The use of the Siegfried map is innovative and provides a good source of historical spatial data, but the temporal spread of the maps across more than four decades introduces a source of temporal inconsistency that is only partially acknowledged. The assumption that these maps provide a coherent "snapshot" of late 19th-century conditions should be supported by more quantitative analysis or stratification. For instance, stratifying map sheets by decade or analyzing whether certain catchments are represented earlier or later in the mapping process could reveal biases in the spatial representation of features. Alternatively, some form of uncertainty band or temporal metadata analysis could be introduced to account for potential inconsistency.

**1) Reply**
The large temporal spread of four decades was due to the presence of two newer map sheets used to complete the area of the Moesa and Calancasca catchments (see Figure 1 below). Without these two map sheets the total temporal spread total for the 10 catchments together is 26 years (from 1872 to 1898).

The temporal spread for the catchments is:

| Catchments | Temporal spread | Years |
|---|---|---|
| Neker and Glatt | 8 | 1879 - 1886 |
| Aabach and Suhre | 13 | 1878 - 1891 |
| Suze | 4 | 1875 - 1879 |
| Venoge | 8 (without 2 marginal older sheets – see Figure 2) | 1890 - 1898 |
| Moesa and Calancasca | 3 (without 2 marginal newer sheets – see Figure 1) | 1872 - 1875 |
| Mattrevispa and Navisence | 12 | 1880 - 1892 |

We have reformulated the paragraph starting in line 92 in this way:

"To cover the entire catchment area of the ten study catchments, we relied on the oldest available map sheets for each catchment, resulting in a time span of approximately 26 years between the oldest (dated 1872) and the newest map sheet (dated 1898). Please note that two newer map sheets dated 1990 and 1915 were employed to cover a very limited area of the Moesa and Calancasca catchments. While the 26 years' time span may seem extensive, it is crucial to note that not all map sheets were regularly updated, and updates were carried out in no apparent order (Räth et al., 2023). Consequently, landscape modifications might only appear on updated map sheets years after they occurred. Thus, we interpret the information from these various map sheets as providing an averaged snapshot of the spatial distribution of surface connectivity in the different catchments during the late 19th century. Furthermore, to the best of our knowledge, no major human interventions or extreme natural events (such as landslides and floods) capable of altering the topography of landscapes and streams have affected the ten study catchments during the period considered."

As noted in line 95, landscape changes may not appear on updated map sheets until years after they occurred. An example is a concrete dam built around 1892 (the "Eberleweiher", in the Glatt catchment), which does not appear on the maps until the 1900 update. Because of these delays, a temporal analysis based on the year of the map sheets would rely on inconsistent timeframes, making it dependent on arbitrary and subjective decisions rather than accurately reflecting when changes actually took place.

[Figure]

**Figure 1**

[Figure]

**Figure 2**

2) (I) The derivation of hydromorphological metrics is clear and well justified, yet the redundancy observed between certain metrics, notably the perfect correlation between drainage and shoreline density, suggests a need for dimensionality reduction or orthogonality testing. Methods such as principal component analysis or hierarchical clustering could be used to examine the independence of metrics and reduce redundancy in the dataset. (II) The use of stream order (Strahler) is appropriate, but it would benefit from a sensitivity test given the known limitations of applying stream order logic to historical maps where ephemeral streams might be underrepresented and headwaters potentially truncated due to mapping resolution or classification ambiguity.

**2) Reply**

(I) Prior to analysis, the non-parametric Spearman's rank correlation test rho (ρ; α = 0.01) was applied to test the occurrence of monotonic relationships among the metrics and landscape variables. When ρ = ± 0.90 only one variable was retained for further analyses. In the case described by the reviewer, we indeed retained only one variable (i.e., drainage density) for further analysis.

(II) Thank you for raising this important point. We acknowledge the limitations of using Strahler stream order, particularly in historical maps where ephemeral streams may be underrepresented and headwaters potentially truncated due to mapping resolution and classification ambiguities. However, after discussion with our statistician, we came to the conclusion that conducting a sensitivity analysis in this context would require us to make a series of subjective assumptions, such as defining what constitutes an ephemeral stream, estimating their likely locations, and determining the extent of headwater truncation. We believe that these assumptions would introduce significant uncertainty and could bias the results rather than improve their robustness.
Given the constraints of historical map data, we opted to work with the available mapped stream network as consistently as possible, recognizing that any attempt to artificially modify stream order would be speculative.

3) There is limited quantitative treatment of uncertainty in the metric derivation process. While the authors recognize spatial inaccuracies in the historical maps and the potential for scanning and digitization errors, these are discussed only qualitatively. A formal uncertainty propagation analysis—perhaps using Monte Carlo simulations that randomly shift hydrographic features within known error

bounds—would provide confidence intervals for key metrics like drainage density and confluence density. Without this, small observed differences between catchments or sub-catchments may not be statistically meaningful.

**3) Reply:**

Thank you for your insightful suggestion regarding uncertainty quantification. The extraction of hydrographic features from historical maps in our study is performed using machine learning algorithms, followed by manual correction. We clarified this in the paragraph starting in line 84, which now reads "The following features were extracted from the scanned map through machine-learning algorithms and translated into digital vector data (see Wu et al., 2022 for more details): the hydrographic network (rivers, streams, lakes and wetlands), the forested areas, the area of the glaciers and the buildings' footprint (i.e. the area on the map covered by the single buildings; Heitzler and Hurni, 2020). All the obtained digital vector data have been subsequently manually corrected to ensure that the extracted vector data had a one-to-one correspondence with what depicted on the map."

While the initial model introduces some uncertainty, the subsequent manual correction ensures that digitization errors are minimized, effectively eliminating defined error bounds for the extracted features. The primary source of uncertainty, therefore, lies in the original historical maps themselves— their accuracy, resolution, and potential inconsistencies in how hydrographic features were represented at the time of their creation [we discuss this in lines 80 to 83].

While we acknowledge this uncertainty, testing its impact by artificially shifting hydrographic features raises several challenges. First, given that we have the valley bottom delineated, streams could theoretically be placed anywhere within this area, making any adjustment inherently subjective. Second, a Monte Carlo approach would require not only shifting the hydrographic features but also adjusting all spatially dependent landscape variables accordingly. This level of spatial dependency makes it difficult to define a realistic and coherent set of assumptions for the randomization process, potentially introducing more uncertainty rather than clarifying the true variability in our metrics.

4) The classification of landscape features from the historical maps, particularly land use and land cover types such as forests, wetlands, buildings, and fields, appears to rely on machine learning, but little detail is given on the classification model used or its validation accuracy. Accuracy assessment using known control areas or expert interpretation would strengthen the validity of the LULC dataset. This is especially important given that key explanatory variables in the regression models depend heavily on this classification. Misclassification of small but hydrologically important features such as wetlands could lead to erroneous model outputs or misinterpretation of connectivity relationships.

**4) Reply:**

Similar to our response to question 3, all hydrological and landscape features extracted from the Siegfried map were manually corrected after processing with machine learning algorithms and vectorization. This thorough verification ensures that misclassification is minimized to nearly zero.

5) The multiscale approach is a major strength of the paper, as it acknowledges the hierarchical nature of hydrological systems. The use of mixed-effects models with nested spatial structure and the control of spatial autocorrelation using locational variables is appropriate and well-motivated. (I) However, no diagnostic tests are reported to assess whether spatial autocorrelation remains in model residuals. Simple tests such as Moran's I or residual correlograms could be used to confirm that spatial dependence has been sufficiently removed, which is crucial for the validity of regression coefficients. (II) In addition, the model structure includes a large number of potential explanatory variables, and while multicollinearity is addressed through VIF analysis, the possibility of overfitting or omitted variable bias remains. Using penalized regression methods or cross-validation could help address this.

**5) Reply:**
(I) As suggested by the reviewer, we carried out Moran's I test on the residuals of the mixed-effect models. The results show that in some of the models, spatial autocorrelation could still be present in the residuals. However, despite being statistically significant, the obtained Moran's I statistics for the residuals across the models are consistently low (average = 0.00479, standard deviation = 0.00514). This suggests that the models have successfully removed most of the spatial dependency present in the data.

(II) We acknowledge that the model structure includes several potential explanatory variables and, despite we have addressed multicollinearity through VIF analysis, there could be the concern regarding overfitting and omitted variable bias. However, the primary goal of using the regression models in this study is to explore and interpret patterns and trends in studied catchments, rather than to identify a highly optimized model. As such, we deliberately retained as many variables as possible, as all are believed to potentially influence the response variable.

6) The assumption that topography, especially the valley bottom extent and slope, has remained unchanged since the 19th century underlies several parts of the validation and metric alignment. While reasonable in many alpine regions, this assumption may not hold in dynamic sedimentary basins or in areas with known anthropogenic modifications such as channelization or urban expansion. In such cases, the comparison between historical hydrography and contemporary DEM-derived valley bottoms may introduce bias. Some discussion or sensitivity testing around topographic stability over the study period would enhance the reliability of the validation procedure.

**6) Reply:**
We employed the contemporary DEM to derive the following variables: (sub)catchments' area, mean elevation, mean slope, percentage of area covered by the valley bottom, and mean valley width. Although these variables may be affected by the dynamics in sedimentary basins such as sediment deposition or river incision, in lack of extreme events these processes take a long time to cause noticeable topology changes. Anthropogenic modifications such as channelization or urban expansion can substantially modify the internal structure of the valleys but should not drastically change the general topology of the valley in terms of width and area.

7) The regression framework is sound and includes both fixed and mixed-effect models, with appropriate use of generalized models and two-part modeling for zero-inflated variables. However, the relationships between hydrological metrics and landscape variables are often complex and potentially nonlinear. The current linear framework may not fully capture these complexities. Generalized additive models (GAMs) could be employed to allow for more flexible, non-parametric relationships between predictors and response variables. This may be particularly useful for variables such as stream order or wetland connectivity, which may exhibit threshold or saturation behaviors.

**7) Reply:**
Thank you for your suggestion regarding the use of GAMs. We recognize that hydrological and landscape relationships can be complex and potentially nonlinear. We also discussed this important point with our statistician. However, applying GAMs effectively typically requires a large and high-quality dataset to robustly estimate smooth functions, which represents a challenge in our case. Our map-derived dataset is inherently noisy, and flexible, data-driven models such as GAMs are prone to overfitting or detecting spurious patterns. Our goal was to employ a model that balances interpretability and complexity—one that is structured enough to account for spatial variables while remaining transparent in its assumptions. By using a simpler regression framework, we aimed to identify broad trends in the data, test our hypotheses, and then validate findings through spatial

analysis on the maps themselves. Introducing more complex models, such as GAMs, could increase the risk of overfitting and drawing inferences that are not strongly supported by the data. Furthermore, even if a GAM revealed a nonlinear relationship, we lack independent, process-based evidence to confirm whether this reflects true ecological mechanisms (e.g., saturation thresholds in wetland connectivity) or is an artifact of data limitations. Without additional field validation, increasing model flexibility may not meaningfully enhance our understanding of the underlying hydrological processes.

We appreciate this suggestion and acknowledge the value of GAMs in certain contexts, but given our dataset's constraints and our study's objectives, we believe our chosen approach provides the most reliable and interpretable insights.

8) While the ecological relevance of hydromorphological metrics is discussed, the connection between these metrics and specific ecological processes or outcomes remains somewhat abstract. The paper could benefit from more explicit hypotheses or examples linking SHC to ecological functions such as habitat continuity, fish migration, or nutrient flux. Even though the historical nature of the data makes direct ecological validation difficult, drawing stronger conceptual links to ecological theory would help bridge the hydrological and ecological dimensions of the study.

**8) Reply:**
We acknowledge the reviewer's suggestion to establish a more explicit connection between SHC and specific ecological functions. While we agree that such links are important, we believe that a detailed exploration of these ecological processes extends beyond the scope of our manuscript. As the reviewer rightly pointed out, the historical nature of our dataset makes direct ecological validation challenging, and any connections drawn between past SHC conditions and past ecological functions present in the studied catchments would be inherently inferential.

Nevertheless, we have reported the ecological relevance of each SHC metric in Table 2. These descriptions provide insight into how different aspects of connectivity may relate to ecological processes such as habitat provision and complexity, transport of nutrients and organisms, species dispersion and recolonization, and network-scale community stability.

Minor comments:

- **Line 12**: "Human activities have progressively altered…"
We decided to retain the sentence as it is, as we think it better connects to the historical framework of the study.

- **Line 23**: "into large-scale hydrological processes"
We corrected he sentence with "…gain a deeper understanding of important hydrological processes at catchment-scale".

- **Line 41**: instead of 'its vegetation cover' "the vegetation cover."
We corrected the sentence as suggested.

- **Line 62**: "Siegfried map,"
The sentence should be correct without the comma.

- **Line 64**: "primary drivers **of** surface hydrological connectivity…"
We corrected the sentence as suggested.

- **Table 1 caption (line 70)**: "Hypothesized relationships between **the** landscape drivers…"
The sentence should we correct without adding "the".

- **Line 88**: "...likely represent the perennial component"

We corrected the sentence as suggested.

- **Line 165**: "Please note that the percentages do not sum up to 100%..." consider placing this in a footnote or table caption to reduce main text clutter.

This information is already part of the table caption and does not appear in the main text.

- Consistency in units and formatting in Table 3 (e.g., km² vs. km2), this applies to other parts in the manuscript.

We uniformed the units as suggested.

- **Line 218**: should be "Figures were generated..."

We corrected the typo.

- **Line 274**: "...to a lesser extent, by the percentage of area covered by buildings..."

We could not understand what the reviewer is suggesting here.

- **Line 286**: "...the average slope mostly along the second axis." → Consider specifying the direction (positive/negative).

To keep consistency within the section, we prefer to not to add this detail here.

- **Table 5**: Ensure consistent use of "Stream order (max)" vs. "Max stream order" and so on across tables and text.

We consistently used stream order (max) in tables and figures and "maximum stream order" in text.

---

## Author Comment (AC2)

**We thank the Reviewer for highlighting these crucial points.**

**Please find our detailed responses below.**

**Reviewer #2:**

This paper by Antonelli et al. presents the methodology and results of the morphological analysis of hydrographic network derived from historical land use maps in Switzerland, with the purpose of linking catchment geographic descriptors with hydrological connectivity.

While I understand the general framework of the study (eg assessing the hydrological connectivity in the past as a reference to assess how it has been altered by anthropogenic modifications) and its interest for assessment of ecological quality of rivers, I don't really see how the methodology and results presented here contribute to this general framework.

In particular:

1) The concepts of surface hydrological connectivity, and also longitudinal and lateral connectivities are not defined. I would have wished to have at least a correspondence with hydrological processes (eg lateral connectivity = groundwater-river interaction?).

**1) Reply:**
A general definition of hydrological connectivity is presented at the beginning of the introduction, and we expanded on this concept within the first two paragraphs of the introduction. We acknowledge the lack of a clear definition of longitudinal and lateral connectivity within the manuscript: we have added this information in the introduction from line 40, which now reads:

"The spatio-temporal patterns of connectivity within a catchment are strongly driven by the structural configuration of the landscape, including its geological and topographic setting, and the vegetation cover (Jencso et al., 2009; Van Nieuwenhuyse et al., 2011; Wohl et al., 2019). These heterogeneous landscape characteristics represent natural elements of (dis)continuity (Benda et al., 2004a; Poole, 2002), which influence the water movement across different dimensions of the hydrographic network - longitudinal (from headwaters to estuaries), lateral (river to and from floodplains or riparian areas) and vertical (river to and from groundwater) (Brierley et al., 2006; Lexartza-Artza and Wainwright, 2011; Pringle, 2001, Turnbull et al., 2018; Ward, 1989)".

2) The historical maps that were used date from the late XIXth century. At this time, the river morphology was not free of human influences, a lot of rivers were already significantly modified / engineered. How do the authors relate this to the more general framework of the introduction about using historical maps as « references » for river restoration?

**2) Reply:**
Our goal is not to propose a framework about using historical maps as a reference for river restoration. In fact, we never mention the word "reference" in our manuscript, and we are fully aware of the risks of treating historical conditions as a fixed standard for restoration. Instead, in the introduction we wished to emphasize the value of historical sources as providers of contextual information rather than a reference model or restoration target.

Since our study does not aim to reconstruct "reference conditions," the fact that the historical maps depict a landscape already influenced by human activity to some extent does not diminish their value. On the contrary, these maps still offer crucial insights into a period when river morphology was not yet subject to fundamental catchment-scale modification and fragmentation (i.e., dam construction, culverting, draining).

We modified the paragraph in the introduction starting at line 50, which now reads: "In today's highly anthropised landscapes, the primary drivers of fundamental catchment's functions such as connectivity are likely to have shifted from environmental to human factors (Allan, 2004; Allan et al., 2021). Historical information can provide essential insights into the past spatial distribution and primary drivers of connectivity, before the hydrographic networks were subjected to fundamental catchment-scale modification and fragmentation (i.e., dam construction, culverting, draining). The historical perspective is crucial for gaining a holistic view on today's catchments functioning, informing modern restoration and conservation efforts, and assessing the outcome of current actions (Higgs et al., 2014). For instance, historical connectivity patterns can provide contextual information for developing and monitoring river restoration projects (Mould and Fryirs, 2018; Wohl et al., 2015). They can also aid in identifying near-natural water bodies and catchments, thus informing conservation prioritization strategies and ensuring efforts are grounded in the natural range of variability of the landscape (Speed et al., 2016; Grabska-Szwagrzyk et al., 2024)".

3) The metrics that were chosen are very classic / general geomorphology metrics. I don't see how they can be informative on river connectivity (how can Strahler order be informative?) and the authors provide no explanation. I would have expected for example something about dams. This is not discussed at all.

**3) Reply:**
We provide an explanation of how we employ the hydromorphological metrics as a proxy of surface hydrological connectivity in section 2.2 and Table 2.

Since our observations are based on two-dimensional cartographic maps, our analysis of connectivity is inherently tied to the riverscape structure and the interaction between the hydrological network and the surrounding landscape. In this context, (hydro)geomorphological metrics, such as drainage density and confluence density, provide valuable insights into longitudinal connectivity within a catchment. Higher values of these metrics typically indicate areas with greater structural connectivity, as a denser and more interconnected network enhances water movement across the landscape [we added this information in line 104, which reads "Higher values of drainage and confluence density typically indicate areas with greater longitudinal connectivity, as a denser and more interconnected network enhances water movement across the landscape".].

While hydrological connectivity is influenced by geomorphology, it is also shaped by the hierarchical organization of the river network. Although stream order does not directly quantify hydrological connectivity, it serves as an important descriptor of network hierarchy. This is why we found it important to include stream order within the employed metrics.

Concerning the expected discussion about the presence of dams, we did not discuss this aspect because there were not dams in the studied catchments at the time considered in this study. We added this information in line 149, which reads: "No dams or other major obstructions to water flow were present in the ten selected catchments at the time of the mapping".

4) The explanatory factors chosen are very general and calculated at the catchment scale. I would have expected more precise / local factors. In particular I am very surprised by the coarse geology classification, with only 2 classes (permeable, not permeable), which is not justified nor discussed. Geology variability can have a lot of effects in particular on groundwater-river interaction, at local

scales. I find also very suspicious that karst is never mentioned, although the presence of karst plays a great role in river morphology (and I believe karst is present in Switzerland).

**4) Reply:**
This study focuses on large-scale hydrological connectivity, as stated in the abstract and introduction. We now modified the title of the manuscript to better convey this message: "Metric-based analysis of the historical drivers of surface hydrological connectivity at catchment-scale". We also changed "large-scale" to "catchment-scale" within the manuscript.

Because the study focuses on catchment-scale, a refined geology classification with a focus on local scales is overly detailed and not appropriate. Instead, we adopted a classification that aligns with the broader spatial scale of our analyses.

Before starting our analyses, we discussed the geological aspect and a possible classification to use in this study with geology experts. The distinction between permeable and impermeable geology is a standard distinction used by the Federal Office of Environment (FOEN) for the classification of river typology (Schaffner, M., Pfaundler, M., Göggel, W. (2013) Fliessgewässertypisierung der Schweiz. Eine Grundlage für Gewässerbeurteilung und -entwicklung. Umwelt-Wissen 1329, 63). Specifically, FOEN differentiates between siliceous and calcareous geological categories, which provide relevant information about rock permeability: calcareous formations generally exhibit higher permeability, while siliceous formations tend to be less permeable.

When defining the permeable and impermeable categories used in this manuscript, we initially considered a more detailed classification considering all categories present in the used sources (Geological map of Switzerland (1:500000) and Hydrogeological map of Switzerland (1:100000)). This has been reported in Table 4, where we specify the rock types included in each category. This approach ensures that our classification is not arbitrarily simplified but rather derived from an aggregation of detailed geological information suited to our study's scale.

Karst is surely present in Switzerland (two of our study catchments are located in the karstic region of Jura). However, karst is never mentioned in the manuscript because the word "karst" refers to a landscape type, or landform, consisting in a combination of limestone with other components (which are mentioned in Table 4).

5) The results are presented separately for two groups of catchments according to the sca3le of the land use maps (1:25000 and 1:50000). However the effect of the scale on the results is never discussed although these results appear to be different for both groups.

**5) Reply:**
We purposely kept the analyses, results and discussions for the catchments at the two different scales separated because we are aware of the difficulty of directly comparing them. We acknowledge the lack of a discussion on the effect of scale on our observations at the 1:50000 scale compared to the 1:25000 scale. We added this information in the discussion, which reads:

"The lower accuracy and higher cartographic generalization of the features extracted from the map sheets at the 1:50000 scale compared the map sheets at the 1:25000 scale can potentially bring to a general underestimation of the values of the derived metrics in the Alpine catchments. However, the values of the different metrics in the Alpine catchments did not show to be systematically lower than the values of the same metrics in the other catchments (see paragraph 3.1). Despite the primary limitation in detecting smaller features at the 1:50000 scale, the overall structure of the hydrological

network is preserved, and the map sheets at the 1:50000 scale remain a valuable source of information for identifying and comparing catchment-scale hydrological patterns".

6) The paper presents the results of many statistical analyses according to various explanatory factors. Some of these results are obvious, some are more surprising. However there is no or very little interpretation of these results in terms of physics / hydrological processes. Without physical interpretation, the impact of the paper falls short.

**6) Reply:**
Since our data are derived from historical cartographic maps, discussing physical hydrological processes (e.g., infiltration, surface runoff, groundwater recharge, and streamflow generation) in detail would be mostly speculative. We lack historical records of past hydrological conditions that may have influenced the patterns observed in the maps. Moreover, since our study focuses on catchment-scale spatial patterns of surface hydrological connectivity, reliable measures of these processes at catchment scale are still very difficult to obtain. That is why most of the times hydrological models are used to investigate these processes at catchment-scale, yet difficult to validate with field data. We believe our manuscript provides a valuable example of how historical maps can offer insights into past hydrological networks, revealing essential information about historical catchment-scale hydrological properties, including connectivity.

7) The authors mention in the discussion (p 25, l 425-435) that the hydrographic network that was considered already incorporates artificial elements. Then I don't understand the purpose of the study that is presented in the introduction as establishing a "reference" of connectivity (see comment 2).

**7) Reply:**
As pointed out in our reply to comment 2, we did not aim at proposing a framework about using historical maps as a reference for river restoration. We clarified this in the introduction where we now state that the historical perspective can inform modern restoration and conservation efforts and can provide contextual information for developing and monitoring river restoration projects. This is different from promoting for the definition of reference values.

Since our goal is not to identify reference conditions, the hydrographic networks we examine can still offer valuable insights into the past spatial distribution and primary drivers of connectivity, even if they are not entirely pristine. Nevertheless, we focus on networks with relatively low human influence, especially when compared to systems that have been extensively modified by dam construction, and extensive culverting and drainage.

8) Following the same idea, I would have expected a comparison with "connectivity" based on a current map. Why was it not done?

**8) Reply:**
The aim of this manuscript is to explore the historical state of the studied catchments, recognizing the past as valuable in its own. Exploring historical connectivity patterns provides critical insights into the long-term shaping of the riverscape, independent of direct comparison with the present. While a temporal comparison can highlight change, it also risks framing the past solely as a point of reference rather than appreciating its significance as a foundation upon which today's riverscapes have developed.

Obviously, we acknowledge the importance of assessing changes over time, and we are currently developing a separate study specifically focused on locating and quantifying such changes. However, that was not the aim of this manuscript, as we believe that the wealth of information and insights gained from analyzing historical maps deserved a dedicated study.

More technical comments

Tables and Figures in the Results sections are too busy and difficult to read. Particular mention to Table 5 that is completely illegible.

**Reply:**

We understand the reviewer's concern. In preparing the tables, we followed the journal's formatting guidelines for submission (i.e., avoid colored table cells and vertical lines; use horizontal lines only above and below the table, as well as between the header and main body). We are confident that the tables will be more readable once formatted according to the journal's official style in case of publication.